# DETER: Detecting Edited Regions for Deterring Generative Manipulations

## Abstract

Generative AI capabilities have grown substantially in recent years, raising renewed concerns about the potential malicious use of generated data, or "deep fakes." Despite being a longstanding and important research topic, deep fake detection research on most existing datasets has not kept pace with generative AI advancements sufficiently to develop detection technology that can meaningfully alert human users in real-world settings. In this work, we introduce *DETER*, a large-scale dataset for *DETE*cting edited image *R*egions and *deter*ring modern advanced generative manipulations. After a comprehensive study of prior literature, our proposed dataset makes contributions along three main axes: the upgrade on modern manipulations via the state-of-the-art generative models; the mitigation of biased spurious correlations in prior deep fake datasets; and a more unified formulation suitable for various detection models in different granularities. Equipped with *DETER*, we conduct extensive experiments and detailed analysis using our rich annotations and improved benchmark protocols, revealing future directions and the next set of challenges in developing reliable regional fake detection models.

## 1 Introduction

Generative AI models such as StableDiffusion (Rombach et al., 2022) and ChatGPT (OpenAI, 2023) have captured significant attention from both the research community and the general public in recent years, following groundbreaking advances in generative modeling. The booming of those generative AI techniques brings numerous advantages and conveniences but also raises heightened concerns about the potential malicious usage of their generated fake data, especially within the context of identifiable human face images. We posit ourselves in the entire research pipeline of deep fake detection, present an in-depth and comprehensive study, covering *the upstream* SOTA generative models and their applications, *the midstream* existing deep fake datasets, as well as *the downstream* fake detection formulation and models, that motivates us to introduce this novel large-scale fine-grained deep fake detection dataset.

**Growing modern GenAI brings new forgery operations and overlooked harmfulness.** In the upstream generative architecture area, Diffusion Models (DMs) (Sohl-Dickstein et al., 2015; Ho et al., 2020; Song et al., 2020) are replacing Generative Adversarial Nets (GANs) (Goodfellow et al., 2014; Karras et al., 2017; Gal et al., 2022) and become the new state-of-the-art generative models by achieving impressive performance in data generation for images (Rombach et al., 2022; Dhariwal & Nichol, 2021; Ho et al., 2022b; Song et al., 2021; Ramesh et al., 2022; Ho et al., 2022a), audio (Kong et al., 2020; Zhu et al., 2023b; Mittal et al., 2021; Lee & Han, 2021), and videos (Ho et al., 2022c; Singer et al., 2022). Among various direct applications of those deep generative models, image editing plays a key role within the context of deep fake detection. Unlike the vanilla unconditional generation process that maps random Gaussian noises to an implicit real data distribution, image editing requires extra controlling mechanisms on the original generative models. Essentially, the above-mentioned unconditional synthesis (whole image generation) and more fine-grained data editing (usually partial image manipulations) lead to distinguishable detection granularities. In the latter fine-grained application area, both GANs-based (Liu et al., 2023c; Yildirim et al., 2023; Li et al., 2022; Pan et al., 2023) and DMs-based methods (Zhu et al., 2023a; Kim et al., 2022; Liu et al., 2023a; Ruiz et al., 2023; Yang et al., 2024b) continue to share an equal footing.

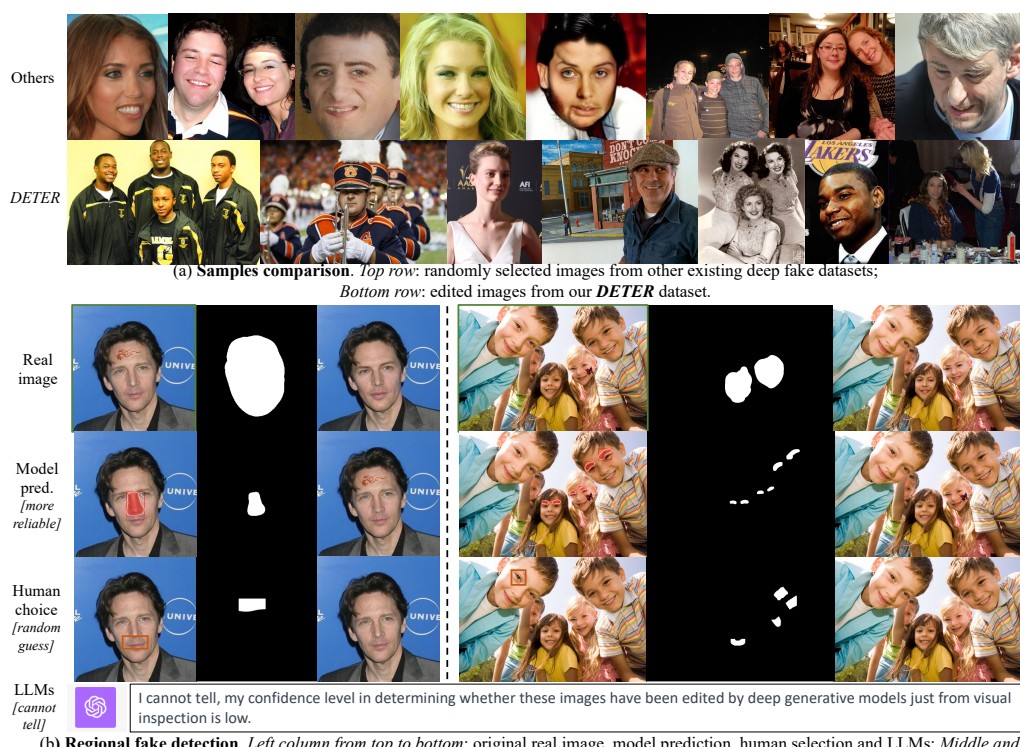

(a) **Samples comparison**. *Top row*: randomly selected images from other existing deep fake datasets; *Bottom row*: edited images from our ***DETER*** dataset.

(b) **Regional fake detection**. *Left column from top to bottom*: original real image, model prediction, human selection and LLMs; *Middle and right columns from top to bottom*: GT masks and their respective *edited images* with face swapping, attribute editing, and inpainting.

Figure 1: **In this work, we introduce the *DETER* dataset for detecting regions manipulated by the state-of-the-art generative models.** By formalizing the problem as a regional detection task, detection models trained on *DETER* can achieve much better performance than human evaluators and popular Large Language Models (LLMs) such as GPT-4 (OpenAI, 2023).

As a closer look at different data editing operations, *face swapping* and *attribute editing* are representative forgery operations adopted in existing fine-grained partially manipulated deep fake datasets (Rössler et al., 2018; Rossler et al., 2019; Li et al., 2020b; Zi et al., 2020; Korshunov & Marcel, 2018; Yang et al., 2019; Dolhansky et al., 2019; Jiang et al., 2020; He et al., 2021; Le et al., 2021), as listed in Tab. 1. However, current generative models can do more than the above. Particularly, a popular branch of recent works can *achieve very photorealistic and natural effects for image inpainting on arbitrary image regions* (Li et al., 2022; Xia et al., 2023; Rombach et al., 2022; Lugmayr et al., 2022), which is a novel type of forgery operations that has not yet been addressed in the detection side. It is worth investigating since it presents a different generation mechanism compared to existing forgery operations. While face swapping and attribute editing rely on the information from reference images to "replace" the target region of unmanipulated images, inpainting techniques leverage the generators' *intrinsic understanding* of the real images to fill in the missing regions. It brings a novel type of risk that has been overlooked in previous literature, as inpainting can change low-level visual information in flexible regions without altering other semantics of the original image such as human identity, as illustrated in the lower-left case of Fig. 1. In a possible real-life scenario, maliciously removing the sign on the face could reverse the person's intentions in public events like a protest.

**Existing datasets for human faces introduce spurious patterns in regional fake detection.** Binary classification formulation (Wang et al., 2023; Corvi et al., 2023; Ricker et al., 2022) where a detector classifies the whole image as being "real" or "fake" is a relatively simplified and idealized situation compared to the real-life malicious scenarios, especially given the emerging versatile applications from the generative front. As an intuitive step forward, OpenForensics (Le et al., 2021) is the first image dataset to introduce fake regional detection and segmentation benchmark tasks. However, despite its efforts to bring the fake detection studies closer to a more fine-grained setup, there is a critical gap to fulfill before building reliable fake detection models: the *spurious correlations* challenge. Specifically, after a deeper investigation into current datasets (Rössler et al., 2018; Rossler

Table 1: **Comparison of basic statistics for regional deep fake datasets.** We list recent popular regional deep fake datasets ordered by time, with their scales, generators and editing operations. Most existing popular deep fake datasets are video-based. Several recent image datasets edit face images with the swapping operation. *DETER* includes the state-of-the-art GANs and DMs-based generators with diverse editing operations and annotations.

| Datasets | Format | Real | Fake | GANs | DMs | FaceSwap | Attribute | Inpaint | Multiple faces | Masks |
|---|---|---|---|---|---|---|---|---|---|---|
| FaceForensics++ 19' (Rossler et al., 2019) | Videos | 1,000 | 4,000 | ✗ | ✗ | ✓ | ✗ | ✗ | ✗ | ✓ |
| Celeb-DF 20' (Li et al., 2020b) | Videos | 590 | 5,639 | ✓ | ✗ | ✓ | ✗ | ✗ | ✗ | ✗ |
| DFFD 20' (Dang et al., 2020) | Images | 1,000 | 3,000 | ✓ | ✗ | ✓ | ✓ | ✗ | ✗ | ✓ |
| DFDC 20' (Dolhansky et al., 2020) | Videos | 23,564 | 104,500 | ✓ | ✗ | ✓ | ✗ | ✗ | ✗ | ✗ |
| ForgeryNet 21' (He et al., 2021) | Videos | 99,630 | 121,617 | ✓ | ✗ | ✓ | ✓ | ✗ | ✓ | ✓ |
| DF-Platter 23' (Narayan et al., 2023) | Videos | 764 | 132,496 | ✓ | ✗ | ✓ | ✗ | ✗ | ✓ | ✗ |
| OpenForensics 21' (Le et al., 2021) | Images | 45,473 | 115,325 | ✓ | ✗ | ✓ | ✗ | ✗ | ✓ | ✓ |
| DGM[4] 23' (Shao et al., 2023) | Images&Texts | 77,426 | 152,574 | ✓ | ✗ | ✓ | ✓ | ✗ | ✗ | ✗ |
| DETER (Ours) | Images | 38,996 | 300,000 | ✓ | ✓ | ✓ | ✓ | ✓ | ✓ | ✓ |

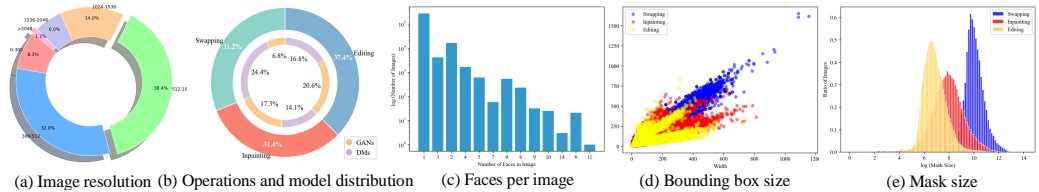

(a) Image resolution   (b) Operations and model distribution    (c) Faces per image    (d) Bounding box size    (e) Mask size

Figure 2: **Statistical distributions in *DETER*.** Our dataset covers images in diverse resolutions, edited via multiple SOTA generators (for this regional manipulation context) with different editing operations and versatile mask sizes and shapes. Best viewed in color with zoom-in.

et al., 2019; Li et al., 2020b; Zi et al., 2020; Korshunov & Marcel, 2018; Yang et al., 2019; Dolhansky et al., 2019; Jiang et al., 2020; He et al., 2021; Le et al., 2021), we note that the detection and segmentation models trained on existing regional fake datasets tend to *capture spurious correlations* during inference, leading to a *high false positive rate* mainly due to the two following reasons.

*Firstly*, face swapping and attribute editing focus on manipulating limited regions of the face (e.g., eyes, nose, and lips). These repetitive patterns cause detection models to frequently predict certain parts of face images as fake regions because these areas are most commonly manipulated in the training data, rather than learning the true generative patterns. To mitigate this, our inpainting operation in *DETER*, which can be deployed on *arbitrary* regions, helps decouple the spurious correlations between certain visual cues (e.g., face shapes) and the forgery operations. *In addition*, the training setup of prior regional deep fake datasets includes only images with at least one fake region, further encouraging the detection model to capture statistical correlations and learn shortcuts from repetitive patterns. To address this, we introduce negative examples (i.e., unmanipulated images) in our improved setup, encouraging the models to accurately detect the true manipulated regions.

**There is an urgent need for a unified evaluation benchmark with strong generalization ability for fake detection at various granularities.** Currently, deep fake detection methods address generative manipulations in separate ways: whole image (Ojha et al., 2023; Yang et al., 2024a), facial region (Lin et al., 2024; Tan et al., 2024a), and flexible region fake detection (Guo et al., 2023; Ma et al., 2023). While these models show promising performance on their respective datasets, generalization across datasets and generators remains critical for real-life deployment. We show in Sec. 4 that *DETER* provides strong generalization across operations, datasets, and generators.

Another important contribution of our work is introducing *a more unified and less biased evaluation benchmark* for detection methods at various granularities. For fine-grained regional detection, we reveal that existing evaluation protocols using classic metrics like Average Precision (AP) *fail to address* the issue of learning repetitive manipulated patterns. Consequently, even basic detection and segmentation models, such as Fast R-CNN (Girshick, 2015) and Mask R-CNN (He et al., 2017), can appear to perform well but often exhibit a high false alarm rate. This is verified and explained in our extensive experiments and breakdown analysis in Sec. 4. For whole image detection methods, our enhanced setup with mixed *negative examples*, along with a newly proposed *region-based image-level classification accuracy* as an additional assessment criterion, supplements standard metrics and evaluates whole fake image classification accuracy in a less biased way. Notably, this approach boosts accuracy and precision by *more than 20%* across various operations and methods.

We believe this work shall help our community build more robust and reliable fake detection systems, with the following main contributions: (1) *DETER* targets new, potentially harmful manipulations enabled by the GenAI age. (2) *DETER* mitigates spurious correlations in prior regional deep fake datasets and improves the experimental setup to encourage detection models to learn true generative patterns. (3) *DETER* provides a unified and comprehensive evaluation benchmark that allows for both whole-image level and fine-grained regional level assessments of existing detection methods.

## 2   RELATED WORK

**Deep Fake Datasets.**   Most existing deep fake datasets can be categorized as either video-based (Rössler et al., 2018; Rossler et al., 2019; Li et al., 2020b; Zi et al., 2020; Korshunov & Marcel, 2018; Yang et al., 2019; Dolhansky et al., 2019; Jiang et al., 2020; He et al., 2021; Dang et al., 2020) or image-based (Le et al., 2021; Shao et al., 2023; Zhou et al., 2017), as summarized in Table 1. All of these datasets provide true or false labels that enable binary classification benchmark tasks, while few of them integrate more fine-grained box or mask-level annotations for fake region detection or segmentation tasks. Face swapping with GANs-based generators is the most commonly adopted forgery operation during the construction, with few including attribute editing. In comparison, *DETER* is the first large-scale dataset that uses the latest state-of-the-art fine-grained methods as generators, and covers editing operations with different granularities. Notably, inpainting is a forgery operation that has never been addressed before in deep fake datasets.

**Generative Models for Image Manipulations.**   While diffusion models (DMs) (Sohl-Dickstein et al., 2015; Ho et al., 2020; Song & Ermon, 2020; Song et al., 2020; Rombach et al., 2022; Ramesh et al., 2022; Dhariwal & Nichol, 2021) are steadily replacing generative adversarial networks (GANs) (Goodfellow et al., 2014; Karras et al., 2017; Gal et al., 2022; Xu et al., 2018) and have become the dominating method for image synthesis in the past two years, GANs have not yet been entirely supplanted in the downstream side for more fine-grained data manipulation applications such as face swapping and inpainting. Among the most recent works that perform fine-grained image manipulations within the past years (Liu et al., 2023c; Yildirim et al., 2023; Li et al., 2022; Pan et al., 2023; Liu et al., 2023b; Zhao et al., 2023; Zhu et al., 2023a; Kwon et al., 2023; Lugmayr et al., 2022; Xia et al., 2023), we carefully select four methods (Liu et al., 2023b; Li et al., 2022; Zhao et al., 2023; Xia et al., 2023) that cover both GANs and DMs backbones based on their editing quality and versatility as the generators in this work.

**Fake Detection Modeling.**   Fake detection methods are closely entangled with available benchmarks and evaluation systems. Many earlier works (Liu et al., 2020; Dang et al., 2020; Li et al., 2020a; Wang et al., 2020; Yu et al., 2019) tackle the problem against GAN-based generators using Convolutional Neural Networks (CNNs) discriminators and can already achieve very high accuracy (more than 99.9%) in discerning fake/real images. Even the most recent fake detection works that build upon diffusion models (Corvi et al., 2023; Ricker et al., 2022; Wang et al., 2023) still follow the conventional setting and formalize it as a binary classification problem. However, the demand for fake detection methods has gone beyond a true or false label, especially given the more sophisticated generators. In this work, we formalize the problem as more fine-grained detection and segmentation tasks.

## 3   DETER FOR FLEXIBLE DEEP FAKE DETECTION IN THE WILD

### 3.1   DATASET OVERVIEW

**Diverse Real-life Scenarios.**   Among different real image datasets that include humans, we select CelebA (Liu et al., 2015) and WiderFace (Yang et al., 2016) as the real human face image sources. The rationales for the above choices is that CelebA (Liu et al., 2015) is one of the most widely adopted datasets in the generative modeling area, and WiderFace (Yang et al., 2016) includes in-the-wild real images that better capture the complex real-life scenarios. Both datasets are open access to the public under proper license (Creative Common License) for non-commercial research purposes.

**Editing Operations.** *DETER* incorporates three image editing operations with varying granularities: face swapping, inpainting, and attribute editing. Specifically, face swapping involves replacing a person's face in a real image with a reference image (face). Inpainting fills in a missing part of an image using generative models without reference images. Attribute editing, similar to face swapping but at a finer grain, involves replacing specific facial regions, such as eyes, ears, and lips. These operations include different editing regions indicated by binary masks. As shown in Fig. 2, their

Figure 3: **Qualitative comparison for different generators on whole image synthesis and regional manipulations with the inpainting operation.** *Upper:* Images generated with the same text prompt *"generate a realistic image of a light-skinned woman walking on the street with a handbag and sunglasses in a purple dress"*. While some large generative models may be SOTA on generic text-to-image generation, it is not difficult for humans to distinguish the fake ones from the real images. *Bottom:* We test various generators that can fulfil the regional editing requirements, such as MAT (Li et al., 2022), DiffIR (Xia et al., 2023), StableDiffusion-v2, (Rombach et al., 2022), SD-XL (Podell et al., 2023), DDNM (Wang et al., 2022), and DiffPIR (Zhu et al., 2023c), and select the ones (MAT and DiffIR) that yield more natural effects for the construction of our dataset.

average editing areas are 31,192, 6,111, and 1,625 pixels, corresponding to squares of 176, 78, and 40 pixels, respectively. While face swapping and attribute editing are common forgery techniques in existing datasets, inpainting is a *unique* feature of *DETER*. Unlike conventional techniques, inpainting does not rely on reference images and can be applied to arbitrary regions, presenting a novel type of forgery that mitigates spurious correlations from previous dataset constructions. Our experimental results in Sec. 4 reveal that, despite having larger editing masks than attribute editing, inpainted regions are more difficult for current models to detect.

**SOTA Generators.** We adopt four state-of-the-art generative models as the deep generators for dataset construction after having extensively examined and compared their editing quality. For *face swapping* and *attribute editing*, we adopt the GANs-based E4S (Liu et al., 2023b) and DMs-based DiffSwap (Zhao et al., 2023); for *inpainting*, we deploy the GANs-based MAT (Li et al., 2022) and DMs-based DiffIR (Xia et al., 2023) as the manipulation tools. Interestingly, while DMs (Ho et al., 2020; Song et al., 2020; Sohl-Dickstein et al., 2015; Ho et al., 2022b) are believed to have surpassed GANs (Goodfellow et al., 2014) in unconditional data synthesis, our analysis suggests the current detection methods are more robust against GANs-based generative techniques, as shown in our cross-generator experiments in Sec. 4. It is important to note that the "state-of-the-art" (SOTA) methods discussed here are defined specifically *within the context of fine-grained regional manipulations*. In other words, while more recent large generative models, such as StableDiffusion 3 (Esser et al., 2024), may be considered SOTA for tasks such as whole-image generation, they may produce less realistic results when applied to localized manipulations, as illustrated in Fig. 3.

**Overall Statistics.** To sum up, *DETER* presents 300,000 edited images based on 38,996 real images. The training, validation, and testing splits are partitioned following the 6:1:3 ratio, which includes 180K, 30K, and 90K edited images, respectively. We incorporate three editing operations via four SOTA generators. Our images cover diverse real-life scenarios that includes both single and multiple faces. Fig. 2 summarizes important statistics about our *DETER* with more details in Appendix A.

## 3.2 CONSTRUCTION APPROACH WITH REPLACEABLE GENERATIVE BACKBONES

Our dataset construction approach, depicted in Fig. 4, and as explained below, can be flexibly adapted to new generative models in a plug-and-play manner, facilitating the need to keep close pace with the fast advancement from the GenAI side.

**Pre-processing.** We first run face detection and alignment methods (Bulat & Tzimiropoulos, 2017) on the real images and parse the detected face to obtain masks with different levels that include the

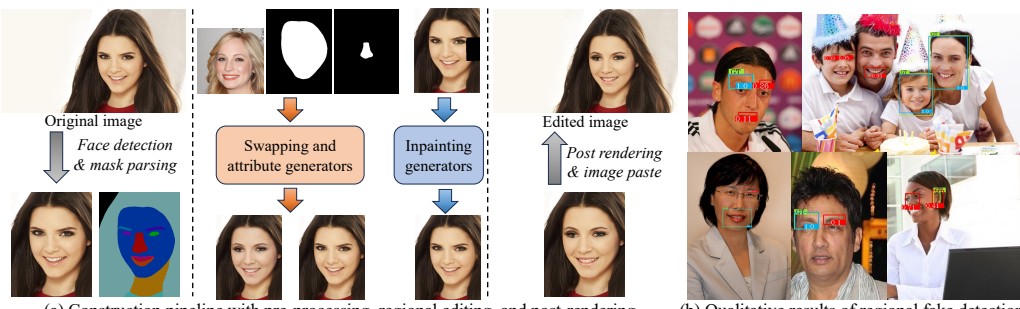

(a) Construction pipeline with pre-processing, regional editing, and post-rendering.    (b) Qualitative results of regional fake detection.

Figure 4: **(a) Pipeline for DETER construction.** Notably, compared to other datasets that require additional conditioning such as labels and prompts during the construction (Shao et al., 2023; Guillaro et al., 2023), our method takes flexible masks as input to generators in a model-agnostic way, facilitating the upgrade of the generative backbones. **(b) Qualitative results of the regional fake detection task.** GT, correct predictions, and false positives are annotated in green, blue, and red boxes, respectively. Existing datasets induce a relatively high false alarm rate. Best viewed in color.

entire face and detailed features such as eyes, lip, and nose (Yu et al., 2018; 2021). The selection of editing masks is based on specific operations. For face swapping and attribute editing, we adopt the face and feature-level masks, respectively. As for inpainting, there are two ways to obtain the editing masks: we either dilate the original feature-level masks into arbitrary shapes or randomly pick an image region within the face mask. The rationale behind our mask generation mechanism for the inpainting operation is to ensure that it has an editing granularity that is between the face swapping and attribute editing, also further decoupling the spurious correlations between low-level face feature characteristics and the editing operations, increasing the difficulties for the model detection as revealed by our experiments in Sec. 4.

**Regional Editing.** After the pre-processing and the mask selection, we then proceed to the editing step. As previously mentioned, we use GANs-based E4S (Liu et al., 2023b) and MAT (Li et al., 2022), and DMs-based DiffSwap (Zhao et al., 2023) and DiffIR (Xia et al., 2023) as the deep generators. Specifically for face swapping and attribute editing, the deep generators also take a reference image as input in addition to the original face image and binary masks. In contrast, inpainting models take an image with missing regions grounded by our editing masks as input and output an image completed by generative models, as shown in Fig. 4 (a).

**Post-processing.** To better ensure the quality of our *DETER* dataset, we apply a series of post-rendering techniques on the output of various deep generators, which include color matching, Poisson fusion, and image sharping. These operations alleviate the boundary effects (i.e., low-level visual image distortions perceivable by human eyes) in conventional forgery construction pipelines and further boost the quality of our dataset. We then paste the face regions back into the original images to get the final edited images, which strictly ensures that our mask annotations precisely reflect the actual regions manipulated by generative models.

**Better Visual Quality and ID Preservation.** We demonstrate the high quality of our dataset in both qualitative and quantitative assessments. As illustrated by the samples in Fig. 1, the edited images from *DETER* can be hardly detected by bare eyes, which is further confirmed by our human studies in the next section. Also, our dataset has a lower ID-distance (Yang et al., 2024b) score of **0.30**, compared to the most recent DGM[4] (Shao et al., 2023) dataset that has the same score of **0.93** based on 10,000 samples, indicating *DETER* has the better identity preservation.

### 3.3 QUALITY ASSESSMENT BY HUMAN STUDY AND LLMS

To validate the visual consistency and fidelity of the proposed dataset, we perform Institutional Review Board (IRB) approved human studies and LLMs-based evaluations with the state-of-the-art GPT-4 from OpenAI (OpenAI, 2023).

While humans are usually believed to be the performance upper-bound in various computer vision tasks to ground the model learning such as object recognition and segmentation (Zhao et al., 2019; Minaee et al., 2021), they become *lower-bound* in fake detection. ChatGPT performs even worse than

Table 2: **User-study and LLMs results for general quality.** "Picks" is the frequency of each type of image selected as the "fake" one. "Detection rate" is the conditional proportion over the selected fake images.

| Choices | Real | Others datasets | *DETER* | Unsure | Total |
|---|---|---|---|---|---|
| Human picks | 38.3% | 23.7% | 15.7% | 22.3% | 100% |
| Human detection rate | - | 60.2% | **39.8%** | - | 100% |
| LLMs picks | 0% | 3% | 2% | 95% | 100% |
| LLMs detection rate | - | 60% | **40%** | - | 100% |

Table 3: **User-study and LLMs results on regional fake selection.** Picking the edited region is a more challenging task for human evaluators, and the rate for picking the GT regions is similar to a random guess.

| Choices | Random regions | GT | Unsure | Total |
|---|---|---|---|---|
| Human picks | 59.0% | 30.3% | 11.7 % | 100% |
| LLMs picks | 0% | 0% | 100% | 100% |

humans in this case, which further confirms the quality of our *DETER*, as well as the great potential and necessity of model assistance when deploying responsible Generative AI in real life.

**General Quality Assessment.** In the first layout, we investigate human performance in general fake detection, which resembles the conventional binary image classification task similar to previous works (Rossler et al., 2019; Liu et al., 2020; Le et al., 2021). Specifically, we prepared 400 image triplets, each including two real images and one edited image, and asked human evaluators to identify the fake one. We also included a supplementary *"I am not sure"* option, allowing evaluators to forfeit instead of forcing a choice on difficult samples. Among the 400 edited images, half were randomly selected from *DETER*, with the other half equally sampled from existing deep fake sources, including SeqDeepFake (Shao et al., 2022), DGM[4] (Shao et al., 2023), OpenForensics (Le et al., 2021), and DDPMs (Ho et al., 2020). The distribution of picks and the detection rate based on correct picks is in Tab. 2. Given an equal population of fake images, the detection rate conditioned on all correct picks on *DETER* is **20.4%** lower than the ensemble of other sources, demonstrating its high quality.

**Regional Fake Detection.** We conduct a more fine-grained layout of human studies for regional fake detection using another 100 triplets. Each triplet comprises the same edited image from *DETER*, with each image grounded in different regions. One region represents the ground truth, while the other two are randomly selected distractors. Similar to the first layout, we ask evaluators to pick the correct region or choose the *"I am not sure"* option. The results in Tab. 3 show that this task is more challenging for humans, with the rate of selecting the ground truth being close to random guessing.

**LLMs Evaluation.** To comprehensively assess the quality of *DETER*, we also use GPT-4 (OpenAI, 2023), a state-of-the-art LLM capable of processing multimodal information, for evaluating fake detection performance. We use proper prompt tuning to set up an evaluation process similar to the human studies for general quality assessment and regional fake detection. The results, based on 100 queries for each evaluation task, are integrated into Tab. 2 and Tab. 3. Our tests show that GPT-4 often gives an uncertain answer, frequently selecting the option *"I am not sure"*.

## 4 BENCHMARK AND ANALYSIS FOR DEEP FAKE DETECTION

### 4.1 IMPROVED EXPERIMENTAL SETUP

As previously mentioned, current evaluation benchmark suites for regional fake detection often introduce spurious correlations during dataset construction, leading to biased and seemingly good performance under conventional task setups and evaluation protocols.

**Conventional and Improved Training Settings.** In traditional object detection and instance segmentation training, models learn to distinguish positive and negative regions *within the same image*. However, relying solely on intrinsic features for self-comparison introduces a strong prior: models tend to assume the presence of target regions in every image. This bias is even further amplified in the case of the regional fake detection problem due to fixed edited region patterns and generators. This contradicts real-life scenarios where many Internet images are unaltered. To this end, we investigate two training settings in our experiments: the conventional setup with no image-level negative samples (i.e., all the training images are from our *DETER* training split, each including at least one positive edited region), and an improved setting with image-level negative samples (i.e., the mixture of our training split and another 140K unseen unmanipulated images).

**Testing Setting Closer to Practice.** The testing setup aligns with our improved training designs, in which we incorporate another 90K unedited images, and each operation task comprises 30K distinct images. This design aims to simulate practical scenarios where a large portion of images is unaltered.

Table 4: **Quantitative evaluation results for regional fake detection under *(C)*onventional (i.e., training w/o negative image samples) and our *(I)*mporved (i.e., training with negative image samples) settings.** We report the scores calculated with IoU=0.5 in the main paper due to space limit, with more results in Appendix C. All metrics are the higher the better; **best** and worst results are marked in **bold** and underlined. Note that the *region-based image-level classification accuracy* is an extra metric in our evaluation protocols that explicitly reflects the *image-level* false alarm rate within the formulation of regional detection and segmentation. *P.* and *R.* denote precision and recall.

| Methods | | Classification (image-level) | | | Object Detection (box-level) | | | | | | | | | Instance Segmentation (mask-level) | | |
|---|---|---|---|---|---|---|---|---|---|---|---|---|---|---|---|---|
| | | Swap | Inpaint | Attribute | Swap | | | Inpaint | | | Attribute | | | Swap | Inpaint | Attribute |
| | Setup | Accuracy | | | P. | R. | AP | P. | R. | AP | P. | R. | AP | Mask AP | | |
| MaskR-CNN 17' | C | 0.51 | 0.43 | 0.41 | 0.25 | **0.97** | **0.97** | 0.24 | 0.92 | 0.86 | 0.35 | 0.95 | 0.87 | **0.96** | 0.85 | 0.87 |
| YOLACT 19' | | 0.52 | 0.45 | 0.45 | 0.08 | 0.97 | 0.96 | 0.06 | 0.89 | 0.77 | 0.10 | 0.91 | 0.77 | 0.96 | 0.75 | 0.77 |
| Mask2Former 22' | | 0.47 | 0.42 | 0.40 | 0.20 | **0.97** | 0.95 | 0.20 | 0.88 | 0.73 | 0.31 | 0.92 | 0.84 | 0.95 | 0.73 | 0.84 |
| FasterR-CNN 15' | | 0.53 | 0.43 | 0.41 | 0.27 | **0.97** | **0.97** | 0.25 | 0.90 | 0.83 | 0.37 | 0.93 | 0.85 | - | - | - |
| YOLOX 21' | | 0.54 | 0.51 | 0.52 | 0.29 | 0.96 | 0.96 | 0.30 | 0.91 | 0.80 | 0.43 | 0.93 | 0.86 | - | - | - |
| DINO 22' | | 0.44 | 0.38 | 0.41 | 0.11 | **0.97** | 0.96 | 0.11 | **0.93** | 0.84 | 0.19 | **0.96** | 0.87 | - | - | - |
| MaskR-CNN 17' | I | 0.75 | 0.68 | 0.64 | 0.45 | **0.97** | 0.96 | 0.41 | 0.91 | **0.88** | 0.53 | 0.93 | 0.89 | 0.96 | **0.88** | **0.89** |
| YOLACT 19' | | 0.85 | 0.78 | 0.74 | 0.47 | **0.97** | 0.96 | 0.35 | 0.88 | 0.85 | 0.45 | 0.88 | 0.83 | 0.96 | 0.83 | 0.83 |
| Mask2Former 22' | | 0.78 | 0.70 | 0.65 | 0.44 | **0.97** | 0.96 | 0.37 | 0.87 | 0.83 | 0.48 | 0.90 | 0.84 | 0.96 | 0.83 | 0.84 |
| FasterR-CNN 15' | | 0.77 | 0.69 | 0.65 | 0.50 | **0.97** | 0.96 | 0.43 | 0.89 | 0.86 | 0.55 | 0.91 | 0.87 | - | - | - |
| YOLOX 21' | | **0.92** | **0.86** | **0.82** | 0.78 | 0.96 | 0.95 | 0.68 | 0.90 | 0.88 | 0.74 | 0.88 | 0.85 | - | - | - |
| DINO 22' | | 0.74 | 0.67 | 0.67 | 0.28 | **0.97** | **0.97** | 0.22 | **0.93** | 0.88 | 0.36 | **0.96** | 0.92 | - | - | - |

**Improved Evaluation Protocols in Different Granularities.** Our evaluation protocols include standard metrics for detection and segmentation tasks, along with an additional *region-based image-level classification accuracy*. This allows detection models to leverage our dataset for methodology development at both whole-image and regional levels. For classic evaluation metrics, we use Precision, Recall, the standard COCO-style Average Precision (AP) for box-level detection, and Segmentation AP for instance segmentation. The *region-based image-level classification accuracy* aims to *reflect the image-level false alarm rate* and reveal box-level false positives, complementing Precision. Models trained in our improved setup predict regions believed to be edited by generative models. During inference, we count detected boxes with an IoU greater than 0.5 with the GT as positive regions. If there are no missed detections or false positives in the image, we consider it correctly classified.

## 4.2 EXPERIMENTAL DETAILS

**Baseline Methods for Generic Regional Detection.** We experiment with six detection and segmentation models covering the most classic to the state-of-the-art methods for the generic regional detection: Mask R-CNN (He et al., 2017), YOLACT (Bolya et al., 2019), Mask2Former (Cheng et al., 2022), Faster R-CNN (Girshick, 2015), YOLOX (Ge et al., 2021), and DINO (Zhang et al., 2023). Among these methods, Mask R-CNN (He et al., 2017) and Faster R-CNN (Girshick, 2015) stand out as well-known convolutional-based two-stage methods, providing a reliable baseline. Mask2Former (Cheng et al., 2022) and DINO (Zhang et al., 2023) build upon the success of DETR (Carion et al., 2020), utilizing the transformer-based architecture to model detection and instance segmentation as a direct set prediction. The remaining methods are single-stage and aim at real-time performance.

**Baseline Methods for Whole Image and Face Region Deep Fake Detection.** We also benchmark additional five deep fake detection methods for both whole image and specific face region fake detection: XceptionNet (Chollet, 2017), ViT (Dosovitskiy et al., 2021), UFD (Ojha et al., 2023), NPR (Tan et al., 2024a), and FreqNet (Tan et al., 2024b). ViT and XceptionNet are two classic methods based on transformer and CNN, respectively, that are widely used in deepfake detection. UFD employs the feature of CLIP as a universal representation for whole image level classification. NPR and FreqNet are the latest methods in the fake image detection field, aiming to capture and characterize generalized structural artifacts and frequency domain learning, respectively.

**Implementation Details.** All the methods used ResNet50 (He et al., 2016) as the backbone for a fair comparison, except for YOLOX (Ge et al., 2021), which utilized DarkNet53 (Redmon & Farhadi, 2018). The models were initialized with COCO pretrained weights to enhance performance. We adhered to default settings with slight modifications in epochs and trained the models on 8 Nvidia RTX 4090. Specifically, for the improved training setting, we do not skip the real images with no forgery regions, but use them as abundant negative samples to update the region proposal networks or classifiers in contrast to the default training where data samples with no foreground bounding boxes usually are skipped.

Table 5: **Quantitative results in terms of deep fake detection methods.** *DETER* can be flexibly adapted for evaluation with conventional deepfake detection methods in different granularities (e.g., binary classification). Note that the Acc. here refers to the classification accuracy.

| Methods | Swap Acc. | Swap AP | Inpaint Acc. | Inpaint AP | Attribute Acc. | Attribute AP |
|---|---|---|---|---|---|---|
| XceptionNet 17' | 71.1 | 71.7 | 62.0 | 58.4 | 56.6 | 65.6 |
| ViT 21' | 64.9 | 55.0 | 57.2 | 50.8 | 49.3 | 59.3 |
| UFD 23' | 60.8 | 79.6 | 55.3 | 56.8 | 53.0 | 57.8 |
| FreqNet 24' | 80.7 | 93.7 | 71.2 | 83.8 | 63.6 | 74.6 |
| NPR 24' | **83.1** | **99.1** | **81.4** | **93.3** | **78.6** | **83.9** |

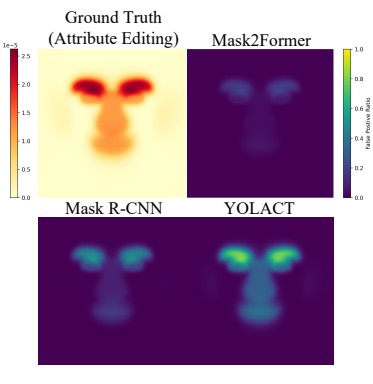

Figure 5: **Distribution of ground truth and false positives for each model in attribute editing task.**

Table 6: **Quantitative results in terms of *(left:)* operations and *(right:)* generators in cross-domain experiments with Mask R-CNN.** Scores calculated with IoU=0.5. Models trained with inpainting data and GANs-based generators achieve better cross-domain performance.

| Train \ Test | Inpaint Precision | Inpaint Recall | AP | Attribute Precision | Attribute Recall | AP |
|---|---|---|---|---|---|---|
| Inpaint | 0.66 | 0.92 | 0.91 | 0.47 | 0.47 | 0.39 |
| Attribute | 0.07 | 0.23 | 0.08 | 0.50 | 0.95 | 0.90 |

| Methods \ Test | GANs Precision | GANs Recall | AP | DMs Precision | DMs Recall | AP |
|---|---|---|---|---|---|---|
| GANs | 0.48 | 0.90 | 0.87 | 0.48 | 0.91 | 0.88 |
| DMs | 0.38 | 0.76 | 0.71 | 0.53 | 0.92 | 0.89 |

## 4.3 EVALUATION RESULTS AND ANALYSIS

We present experimental results and analysis below, with additional details in Appendix C. Note that all the reported results are robust and statistically important with a std in an order of $10^{-3}$.

**Spurious Correlations and Mitigation via Inpainting.** The editing regions for face swapping, attribute editing, and inpainting operations are approximately squares of 176, 78, and 40, respectively. While the detection difficulties are seemingly related to the area of edited regions by intuition, i.e., larger areas of modification tend to be easier to detect, we observe that this does not hold for current detection and segmentation models as shown in Tab. 4. Specifically, we note the edited regions with *inpainting* are consistently more difficult to predict compared to both *face swapping* and *attribute editing*. For example, the precision on inpainting data is on average 0.11 lower (i.e., 0.30 versus 0.41) than that of attribute editing across all models. The operation-wise difficulty variance further validates our initial claim on the spurious correlations introduced in the dataset construction stage with oversimplified editing types. Our proposed *DETER* dataset seeks to mitigate the above by integrating *inpainting* to diversify the editing regions and shapes.

**Extension to Detection Methods in Other Granularity.** Considering most existing deep fake detection methods are designed to classifying entire images, we follow the previous methods and extract the facial regions to form a subset containing real/fake facial images. Tab. 5 lists the results of those methods in whole-image and facial regional granularity on *DETER*. We observe that the accuracy of different methods on the face swapping, inpainting, and attribute editing decreases sequentially, indicating that the modification of region size affects the performance of those detection methods. Notably, the accuracy here refers to whether the current image is classified as real or fake, which is entirely different from the *region-based image-level classification accuracy* in Tab. 4.

**Generalization Ability across Operations.** We conduct cross-domain experiments to study the generalization ability of different editing operations. As shown in the left side of Tab. 6, the model performs much better in in-domain testing (training and testing on the same editing operation) and performs worse in the cross-domain case. We also observe the model trained on the inpainting data has better cross-domain generalization performance compared to the one trained on attribute-edited data. The main reason is that the flexible inpainting operation in *DETER* can be applied on arbitrary face parts, and thus, the model captures the better intrinsic difference between real and manipulated regions, rather than just memorizing the position prior/bias in the training data. As a result, the model

trained on inpainting data has a precision of 0.47 on attribute-edited test samples, similar to the in-domain test precision of 0.50. Our take-away message here is that regional fake detection models should consider the inpainting training data to avoid spurious correlation.

**Generalization Ability across Datasets.** To ensure that *DETER* provides trained detection models with strong generalization abilities, we perform cross dataset experiments with OpenForensics (Le et al., 2021) on face swapping task. When trained with *DETER*, model exhibits strong generalization to OpenForensics, achieving a detection AP of 0.69 during. In contrast, the detection model trained on OpenForensics struggles to detect fake regions in *DETER*, with an AP of 0.02.

**Image-level and Region-level False Alarms.** The comparisons among various metrics further reveal the high false alarm rate across existing detection and segmentation methods. Particularly, the models tend to achieve very high recall (e.g., greater than 0.9) but low precision (e.g., lower than 0.3) in the conventional setup. This recall-precision contrast indicates that the models' predictions involve a large number of real regions that have been predicted as fake, as shown in Fig. 4(b). The same issue is further supported by our *region-based image classification accuracy*, through which we find a lot of real images are classified as edited, resulting in low classification accuracies. This is undesired when deploying a reliable regional fake detection system in practice, where most images on the Internet should still be free of generative manipulations.

**Improved Setup with Negative Samples.** Another dimension of our break-down analysis focuses on the improved task setup with mixed real images in training. Tab. 4 also include the evaluation results obtained under both conventional training and improved training setup. Our improved setup *significantly* boost the classification accuracy and precision by *more than 20%* across operations and methods, demonstrating its effectiveness.

**GANs vs. DMs Generators.** We also conduct cross-domain experiments on the generative models, with results shown in the right side of the Tab. 6. We report the Precision, Recall, and AP scores under the *inpainting* operation task trained with the conventional setting as an illustration example (more generator-based cross-domain results in Appendix C). We observe that detection models trained with the GANs-based generators can generalize well to the DMs-based testing images, while the inverse setting induces a non-trivial performance drop. Our findings suggest that the GANs-based generators include more robust features that are perceivable by detection models.

**Visualization of Error Patterns in Regional Detection.** To delve deeper into the performance of different models on *DETER*, we visualize the probability distributions of ground truth and false positives in the predictions of various models for the attribute editing task in Fig. 5. It can be observed that all models tend to make errors in predicting features such as eyes and eyebrows, with relatively high occurrences in the ground truth. In comparison, false positives generated by Mask2former (Cheng et al., 2022) are generally fewer, while YOLACT (Bolya et al., 2019) yields a considerable number of erroneous predictions.

## 5 DISCUSSION AND CONCLUSION

**Broader Social Impact.** We seek to raise awareness of the potential malicious impact of current GenAI and support future research on building effective and robust detection systems. Necessary safeguards have been adopted while using GenAI techniques for image manipulations to ensure none of sensitive or personally identifiable information is collected during our studies. All of the images in *DETER* are derived from existing public open access datasets under proper license (Creative Common License) for non-commercial research purposes. The human studies and data analysis are conducted under appropriate Institutional Review Board approval and regulations.

**Conclusion and Future Directions.** We introduce our *DETER* dataset for the regional deepfake detection task, featuring a large-scale and high-quality image dataset. We ensure the quality of our benchmark to catch up with the fast-developing generative AI techniques, including SOTA generators, novel forgery operations, deep-dive investigations on current benchmarks and their problematic spurious correlation issues, as well as improved benchmark designs as mitigation. For future research on the detection methods, we explicitly emphasize the significance of a more comprehensive and less biased evaluation system that reflects the real performance of models, with particular attention on the false alarm rate when deployed in real-life scenarios.

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

In the appendices, we provide additional details about our *DETER* dataset in Sec. A. Sec. B describes more details about our human studies. More experimental results and analysis can be found in Sec. C.

# A  MORE DETAILS ABOUT *DETER*

*DETER* includes 300,000 edited images in total, obtained with three editing operations, as described in our main paper. For face swapping, inpainting, and attribute editing, there are 93636, 94253, and 112111 images, which corresponds to 106673, 114066, and 199958 regional manipulation masks, respectively. The image resolutions vary based on the real images, from smaller than 300 to greater than 2048. Fig. 6 and Fig. 7 show the distributions of editing masks and their detailed box heights and widths.

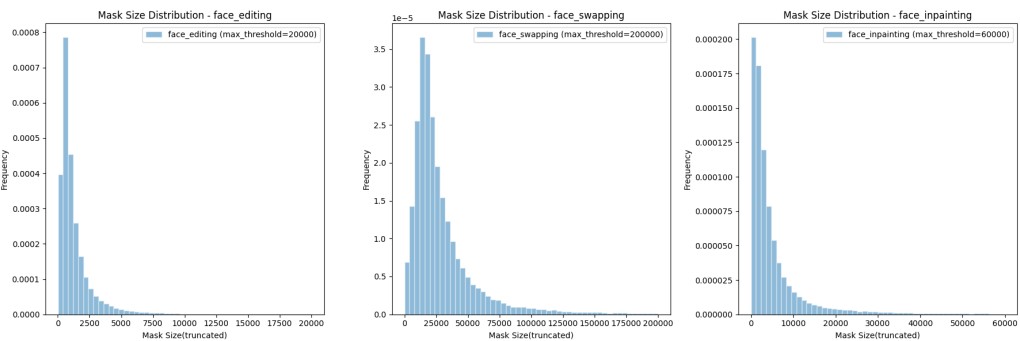

Figure 6: **Distributions of mask sizes in terms of different manipulation operations.**

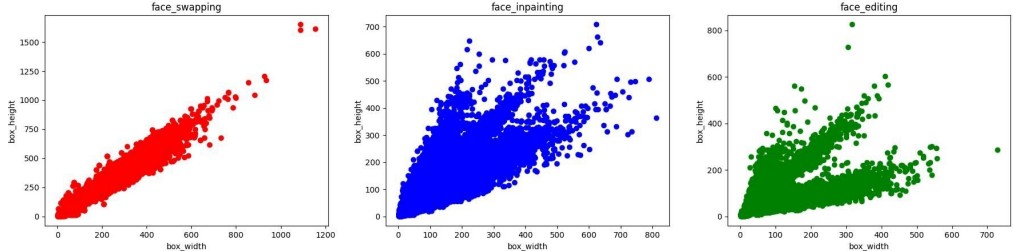

Figure 7: **Details of box sizes.** *Face swapping* operation has the largest average editing area, followed by *inpainting*, and *attribute editing*.

Fig. 8 show more qualitative comparisons between other existing deep fake datasets including DFFD 20' (Dang et al., 2020), SeqDeepFake 22' (Shao et al., 2022), DGM$^4$ 23' (Shao et al., 2023), FaceForensics++ 19' (Rossler et al., 2019), ForgeryNet 21' (He et al., 2021), and OpenForensics 21' (Le et al., 2021).

# B  MORE DETAILS ABOUT HUMAN STUDIES

This section describes further details about our human studies. We organize our human studies in two settings. The first task: **General Quality Assesment** is selecting the fake image from a triplet of 2 real photos and a fake. This task is aimed at evaluating the difficulty of spotting the fake images generated by our method vs other methods used in existing datasets. We use human error in selecting the fake image, as a proxy for the difficulty of spotting cues of deepfake generation, hence the realistic quality of the fake image sample.

The second task: **Regional Fake Detection** is to select the edited region of a photo. We create samples with a specific facial feature/ region of the face edited or altered using our method. Each image triplet for this task involves the same edited image from *DETER* but grounded with different regions, among

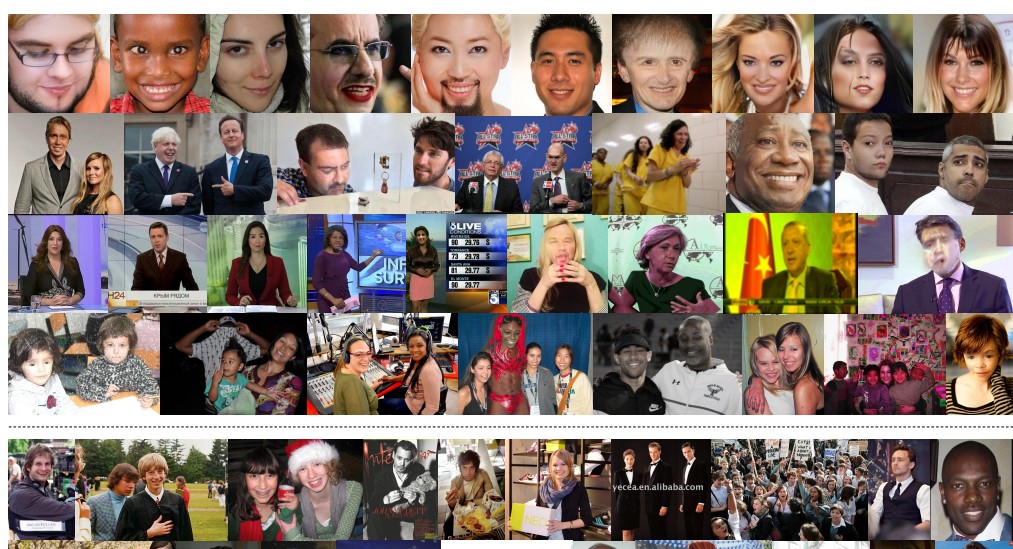

Figure 8: **More qualitative comparisons among samples from different deep fake datasets.** Image samples in *upper* rows come from existing deep fake datasets: DFFD 20' Dang et al. (2020), SeqDeepFake 22' Shao et al. (2022), DGM[4] 23' Shao et al. (2023), FaceForensics++ 19' Rossler et al. (2019), ForgeryNet 21' He et al. (2021), and OpenForensics 21' Le et al. (2021), while the *bottom* rows include samples from our *DETER*.

which one is the ground truth region that has been edited, with the other two untouched regions randomly selected as distractors. We use human error in grounding the edited regions as a proxy for the realistic and subtle nature of localized feature/attribute alterations achieved in our dataset.

### B.1 CROWDSOURCING AND SETUP

We hosted the two evaluation tasks as separate web apps and crowdsourced them through Cloud Research. For the first task, we prepared 400 total image triplets, each including two real images and one edited image. Fake images for 200 of these triplets were randomly selected from our *DETER*, and another 200 equally sampled from existing deep fake sources including SeqDeepFake (Shao et al., 2022), DGM[4] (Shao et al., 2023), OpenForensics (Le et al., 2021), and DDPMs (Ho et al., 2020). For the second task, we had 100 triplets assembled using 100 photos from our dataset with random facial features/regions altered.

For both tasks, in addition to three image options, we also included a *"I am not sure"* option, which allows the evaluators to forfeit instead of forcing them to make a choice when it comes to hard samples. The layout of the survey for one selection is shown in Figures 11 and 12 respectively for tasks 1 and 2.

For both tasks, we split our triples into multiple surveys containing 50 image triplets each. Each survey with 50 image triplets was completed by 3 human evaluators. To ensure that crowdsourced human evaluators spend adequate time looking for cues of deepfakes in each selection, we encourage them to spend at least 20 seconds on each selection. The instructions given to the evaluators for the two tasks are shown in Figure 9 and 10.

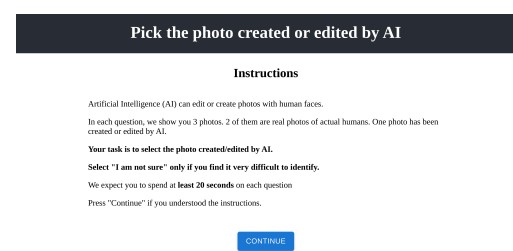

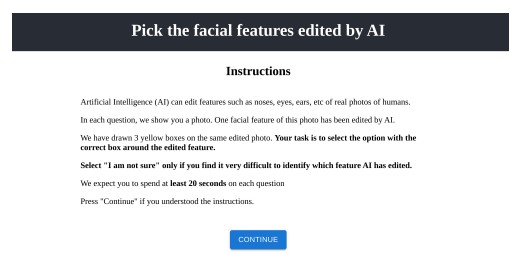

Figure 9: **Task instructions for human evaluation - General Quality Assessment**

Figure 10: **Task instructions for human evaluation - Regional Fake Detection**

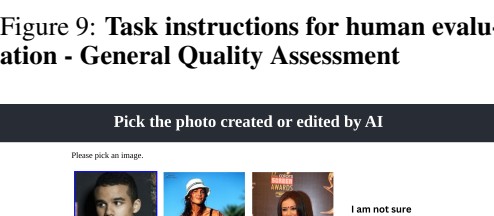

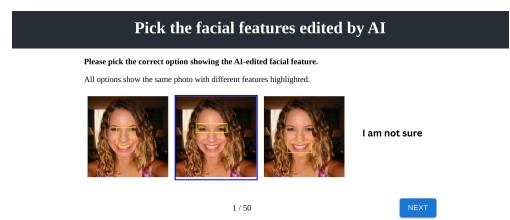

Figure 11: **Task layout for human evaluation - General Quality Assessment**

Figure 12: **Task layout for human evaluation - Regional Fake Detection**

## C MORE DETAILS ABOUT REGIONAL FAKE DETECTION

We report the experimental results measured with IoU=0.5 in the main paper, and provide additional results with IoU=0.75 in Tab. 7. The additional results further validate and support our break-down analysis and take-away messages presented in the main paper.

As shown in Tab. 7, the performance of various models uniformly decreases with the increasing stringency of IoU constraints, aligning with the overall conclusion of the main paper. Specifically, among the three tasks, inpainting exhibits the poorest performance. This is primarily attributed to the arbitrary shape of masks, in contrast to the relatively fixed mask transformation ranges in the other tasks, further underscoring the issue of spurious correlations during the dataset construction stage. In attribute editing, the modified regions are more fixed compared to face swapping, focusing on specific facial areas such as the eyes, mouth, and nose. Consequently, attribute editing achieves the highest precision. Despite its elevated recall, the precision across all tasks remains at a relatively low level. This discrepancy indicates that the model has biases in the learning process, where it fails to adequately capture the inherent differences between features in real and fake images, leading to a significant number of false positives. To address this issue, we introduce additional real images, i.e., improved settings in Tab. 7, during the training process to encourage the model to better discern between real and fake images. This strategy results in a substantial improvement of over 20% in precision and accuracy across all tasks and methods. Therefore, ensuring comprehensive learning of distinctions in features between real and fake images is a crucial focal point for advancing the task of fake regional detection. Fig. 13 includes more qualitative samples.

We have also provided additional generator-based cross-domain results for both inpainting and attribute editing tasks. From Tab. 8, it is evident that the cross-domain performance of models trained with the GANs-based generators significantly surpasses those trained with the DMs-based generators, even outperforming the original DMs domain in inpainting tasks. Specifically, models trained with GANs-based generators exhibit superior performance on GANs-based and DMs-based test data (GANs + DMs), once again highlighting the robustness of features generated by GANs over DMs-based features. Additionally, there is complementary information in the features of GANs-based and DMs-based generators, and joint training further enriches the representation of fake features, leading to better results.

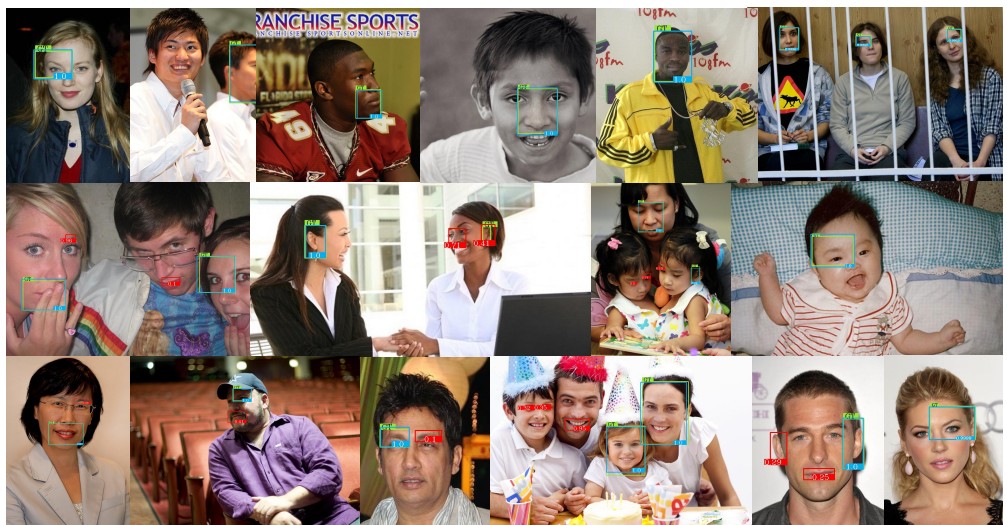

Figure 13: **Additional qualitative results of regional fake detection.** GT, correct predictions, and false positives are annotated in green, blue, and red boxes, respectively. Current models induce a relatively high false alarm rate.

Table 7: **Quantitative evaluation results for regional fake detection under *(C)*onventional (i.e., training w/o negative image samples) and *(I)*mporved (i.e., training with negative image samples) settings with IoU=0.75.** All metrics are the higher the better, best and worst results are marked in **bold** and underlined, respectively.

| Methods | | Classification (image-level) | | | Object Detection (box-level) | | | | | | | | | | Instance Segmentation (mask-level) | | |
|---|---|---|---|---|---|---|---|---|---|---|---|---|---|---|---|---|---|
| | Operations | Swap | Inpaint | Attribute | Swap | | | Inpaint | | | Attribute | | | Swap | Inpaint | Attribute |
| | Setup | Accuracy | | | Precision | Recall | AP | Precision | Recall | AP | Precision | Recall | AP | Mask AP | | |
| MaskR-CNN 17' | C. | 0.51 | 0.40 | 0.40 | 0.24 | **0.97** | **0.96** | 0.20 | 0.77 | 0.73 | 0.33 | 0.88 | 0.81 | 0.958 | 0.718 | 0.807 |
| | I. | 0.75 | 0.65 | 0.62 | 0.45 | 0.96 | 0.95 | 0.35 | 0.77 | 0.74 | 0.49 | 0.86 | 0.82 | 0.954 | **0.738** | **0.818** |
| YOLACT 19' | C. | 0.52 | 0.41 | 0.42 | 0.08 | 0.96 | **0.96** | 0.05 | 0.67 | 0.60 | 0.08 | 0.78 | 0.68 | 0.955 | 0.564 | 0.655 |
| | I. | 0.85 | 0.73 | 0.71 | 0.46 | 0.96 | **0.96** | 0.27 | 0.69 | 0.64 | 0.39 | 0.76 | 0.70 | **0.959** | 0.617 | 0.685 |
| Mask2Former 22' | C. | 0.47 | 0.38 | 0.38 | 0.20 | 0.96 | **0.94** | 0.16 | 0.70 | 0.56 | 0.28 | 0.85 | 0.76 | 0.946 | 0.578 | 0.758 |
| | I. | 0.78 | 0.65 | 0.63 | 0.44 | 0.96 | 0.95 | 0.28 | 0.67 | 0.61 | 0.44 | 0.82 | 0.75 | 0.953 | 0.638 | 0.735 |
| FasterR-CNN 15' | C. | 0.53 | 0.40 | 0.39 | 0.27 | **0.97** | **0.96** | 0.20 | 0.72 | 0.67 | 0.33 | 0.84 | 0.78 | - | - | - |
| | I. | 0.77 | 0.66 | 0.63 | 0.50 | 0.96 | 0.95 | 0.35 | 0.73 | 0.69 | 0.50 | 0.83 | 0.78 | - | - | - |
| YOLOX 21' | C. | 0.54 | 0.48 | 0.49 | 0.29 | 0.96 | 0.95 | 0.26 | 0.77 | 0.69 | 0.40 | 0.86 | 0.80 | - | - | - |
| | I. | **0.92** | **0.82** | **0.79** | **0.77** | 0.95 | 0.95 | **0.58** | 0.77 | 0.74 | **0.69** | 0.81 | 0.79 | - | - | - |
| DINO 22' | C. | 0.44 | 0.36 | 0.40 | 0.11 | **0.97** | **0.96** | 0.09 | 0.78 | 0.72 | 0.18 | **0.90** | 0.82 | - | - | - |
| | I. | 0.74 | 0.65 | 0.65 | 0.28 | **0.97** | **0.96** | 0.19 | **0.79** | **0.75** | 0.33 | **0.90** | **0.85** | - | - | - |

Table 8: **Quantitative results in terms of GANs-based and DMs-based generators in cross-domain experiments with Mask R-CNN (He et al., 2017).** The scores are calculated with IoU=0.5.

| | GANs | | | | | | DMs | | | | | | GANs + DMs | | | | | |
|---|---|---|---|---|---|---|---|---|---|---|---|---|---|---|---|---|---|---|
| | Inpaint | | | Attribute | | | Inpaint | | | Attribute | | | Inpaint | | | Attribute | | |
| | Precision | Recall | AP | Precision | Recall | AP | Precision | Recall | AP | Precision | Recall | AP | Precision | Recall | AP | Precision | Recall | AP |
| GANs | 0.48 | 0.9 | 0.87 | 0.56 | 0.94 | 0.91 | 0.48 | 0.91 | 0.88 | 0.42 | 0.79 | 0.67 | 0.48 | 0.91 | 0.87 | 0.50 | 0.88 | 0.82 |
| DMs | 0.38 | 0.76 | 0.71 | 0.43 | 0.78 | 0.64 | 0.53 | 0.92 | 0.89 | 0.59 | 0.95 | 0.92 | 0.44 | 0.83 | 0.79 | 0.50 | 0.85 | 0.77 |
| GANs + DMs | 0.52 | 0.91 | 0.88 | 0.58 | 0.95 | 0.91 | 0.55 | 0.93 | 0.91 | 0.59 | 0.95 | 0.92 | 0.53 | 0.92 | 0.89 | 0.58 | 0.95 | 0.91 |

