# OpenReview forum: "DETER: Detecting Edited Regions for Deterring Generative Manipulations"
_ICLR.cc/2025/Conference — Submitted to ICLR 2025_

### Official Review · Reviewer_4QTU · 2024-11-03

**Soundness:** 3
**Presentation:** 2
**Contribution:** 3
**Rating:** 5
**Confidence:** 4

**Summary:**

The paper presents an image dataset for regional deepfake detection task. Recent images generated by current generative models are included.

I have read the response of the authors and the comments of other reviewers. I would keep my original score.

**Strengths:**

1. The research problem is useful.
2. Dataset for regional deepfake detection is indeed valuable for deep fake detection tasks.
3. The organization of the paper is good.

**Weaknesses:**

1. The size of the dataset is not large, which I am afraid cannot be termed as 'large-scale'.
2. In section 4.2, method DINO (Zhang et al., 2023) is published in 2023, in Table 4, it is referred to as DINO' 22.
3. In Table 4, the methods are relatively not up-to-date.
4. Grammar errors:
a. we use GANs-based E4S (Liu et al., 2023b) and MAT (Li et al., 2022),
and DMs-based DiffSwap (Zhao et al., 2023) and DiffIR (Xia et al., 2023) as the deep generators.
b. Too long sentence: We posit ourselves in the entire research pipeline of
deep fake detection, present an in-depth and comprehensive study, covering the upstream SOTA
generative models and their applications, the midstream existing deep fake datasets, as well as
the downstream fake detection formulation and models, that motivates us to introduce this novel
large-scale fine-grained deep fake detection dataset.

**Questions:**

See Weaknesses

---

> ### Author Response · Authors · 2024-11-22
>
> > **Q1:** The size of the dataset is not large, which I am afraid cannot be termed as 'large-scale'.
>
> **A1:** As shown in Table 1, our dataset contains significantly more fake images (300,000) than existing datasets, such as DFFD (3,000), OpenForensics (115,325), and DGM (152,574), even though OpenForensics is claimed to be a large-scale dataset (the title of the work is "OpenForensics: Large-Scale Challenging Dataset For Multi-Face Forgery Detection And Segmentation In-The-Wild").
> Thus, we believe it is justifiable to claim that our dataset is in fact large-scale in the domain of regional fake image detection.
>
> ---
>
> > **Q2:** In section 4.2, method DINO (Zhang et al., 2023) is published in 2023, in Table 4, it is referred to as DINO 22'.
>
> **A2:** Thank you for pointing this out. We will correct it to DINO 23' to keep consistent following the suggestion.
>
> ---
>
> > **Q3:** In Table 4, the methods are relatively not up-to-date.
>
> **A3:** We selected a range of classic and representative methods for regional detection and segmentation, as shown in Table 4, which are widely used in both academia and industry. However, despite the significant performance differences across these methods on the classic detection dataset COCO, their performance on our dataset is relatively minor. Therefore, there is no urgent need to use some up-to-date methods.
>
> ---
>
> > **Q4:** Grammar errors and long sentences.
>
> **A4:** Thank you for the careful reading and suggestions. We will correct the grammar and shorten the mentioned sentences accordingly.

---

### Official Review · Reviewer_V9S3 · 2024-11-03

**Soundness:** 3
**Presentation:** 3
**Contribution:** 3
**Rating:** 5
**Confidence:** 4

**Summary:**

This paper introduces a dataset for detecting AI-generated image manipulations. There are 3 main contributions:
1. A new dataset that contains state-of-the-art generation methods, which are not incorporated in previous datasets
1. An improved evaluation framework that considers both whole-image and region-specific manipulations
1. Provides techniques to mitigate spurious correlations in fake detection

The dataset contains 300,000 edited images using three types of manipulations: face swapping, attribute editing, and inpainting, created using both GAN and DM-based generators.

**Strengths:**

The new dataset introduced by this paper can potentially make valuable contributions by addressing a critical challenge in the era of widespread AI image manipulation. It has the following merits:
- Incorporates current state-of-the-art generative models
- Comprehensive evaluation framework
- Extensive experiments with multiple detection methods
- Well-documented human evaluation studies
- Thorough cross-domain analysis

Clarity. This paper has clear methodology presentation and well-structured experimental results

**Weaknesses:**

The main concerns about this paper are ethical and legal concerns.
- There is no clear discussion of whether they have rights to modify/redistribute CelebA and WiderFace images. For example, the CelebA dataset has the following agreement: “You agree not to further copy, publish or distribute any portion of the CelebA dataset. Except, for internal use at a single site within the same organization it is allowed to make copies of the dataset.”
- And there is no discussion of consent from individuals in the images.
- And there is no discussion of rights clearance for redistributing modified faces.

**Questions:**

1. How do you ensure the dataset remains relevant as new generative models emerge?
1. How does the post-processing pipeline affect the natural artifacts that might help in detection?
1. Why not include more diverse manipulation techniques beyond facial modifications?

**Details Of Ethics Concerns:**

1. No clear discussion of whether they have rights to modify/redistribute CelebA and WiderFace images.
1. No mention of original image licenses or usage restrictions.
1. No discussion of consent from individuals in the images
1. Unclear if celebrity images can be legally manipulated and redistributed.

---

> ### Author Response · Authors · 2024-11-22
>
> > **Q1**: (a) There is no clear discussion of whether they have rights to modify/redistribute CelebA and WiderFace images. For example, the CelebA dataset has the following agreement: “You agree not to further copy, publish or distribute any portion of the CelebA dataset. Except, for internal use at a single site within the same organization it is allowed to make copies of the dataset.” (b) And there is no discussion of consent from individuals in the images.
> > (c) And there is no discussion of rights clearance for redistributing modified faces.
>
> **A1:** Thank you for raising these important points regarding licensing.
> The WiderFace dataset is distributed under a Creative Commons (CC) license, as mentioned in Lines 210 and 530 of our paper. According to the terms of the CC license, the dataset can be reproduced and distributed for non-commercial and research purposes, which aligns with the goals of our work.
>
> For the CelebA dataset, as carefully pointed out by the reviewer, we acknowledge that this statement precludes redistribution. To address this concern, we propose a solution that ensures compliance with the dataset's terms. Specifically, instead of redistributing the original images, we will provide only the masks and our construction and evaluation scripts. This approach allows researchers to reproduce our work using the CelebA dataset under their own institutional access while adhering to the licensing agreement. We believe this solution balances open science and legal compliance. If you have any other suggestions, please let us know, we want to make sure we do the right thing here.
>
> While the CelebA and WiderFace datasets are publicly available and widely used in academic research, we acknowledge that they do not explicitly address consent from individuals depicted in the images. Our work strictly adheres to the datasets' licensing terms and is conducted solely for academic purposes. Additionally, we plan to distribute only masks and construction scripts, not the original images, to minimize ethical concerns while ensuring reproducibility.and we appreciate the reviewer’s input as part of this broader conversation in the research community.
>
> Our work will follow the standard Creative Commons license for non-commercial research use, which we will further clarify in the paper and upon the release.
>
> ---
>
> > **Q2:** How do you ensure the dataset remains relevant as new generative models emerge?
>
> **A2:**
>
> Our dataset construction pipeline is designed to be flexible and adaptable, making it straightforward to incorporate newer generative models as they emerge. If more powerful models become available, they can be easily integrated into the pipeline to enhance the dataset’s diversity and utility.
>
> We will release the construction code as part of our work. We appreciate this valuable feedback and agree on the importance of keeping updated with advancements in generative modeling. As part of our maintenance plan, we will regularly update the testing set to reflect emerging methods, encouraging the generalizability of detection models trained on DETER against more diverse generators.
>
> ---
>
> > **Q3:** How does the post-processing pipeline affect the natural artifacts that might help in detection?
>
> **A3:** The post-processing pipeline employs computer graphics techniques such as color matching, Poisson blending, and image sharpening to harmonize the colors between the generated region and the original image. It also smooths the boundary where the generated region meets the original image, thereby reducing visible artifacts that can be detected by the detection model.
>
> ---
>
> > **Q4:** Why not include more diverse manipulation techniques beyond facial modifications?
>
> **A4:** Thank you for your question. In this work, we focus on datasets involving human faces because the manipulation of human images, particularly faces, poses some of the most harmful and urgent societal implications. By targeting this critical area, we aim to address one of the most pressing challenges in the realm of generative model misuse and detection.
>
> However, our proposed construction method has strong generalization capability and can be readily applied to other scenarios, such as natural scenes or animals. As described in Section 3.2, our dataset construction pipeline includes pre-processing, regional editing, and post-rendering, all of which are adaptable to different contexts. For instance, pre-processing can involve specifying regions to be edited, either manually or using segmentation methods like SAM. Regional editing can be performed using generative models tailored to specific scenes, and the post-processing step is not constrained by the input's nature.
>
> To support broader use cases, we will release our construction code, enabling the research community to extend the approach to manipulation techniques beyond human faces.
>
> ---
>
> **Other ethical concerns**: Please refer to our **A1** for detailed responses.

---

### Official Review · Reviewer_KjPK · 2024-11-04

**Soundness:** 3
**Presentation:** 3
**Contribution:** 3
**Rating:** 8
**Confidence:** 3

**Summary:**

This paper proposes DETER, a large-scale dataset for detecting edited images and regions. The contributions of this work include: 1) in addition to face swapping and attribute editing, the authors also include image inpainting in the data editing operations; 2) to address the spurious correlation challenge in the current datasets for regional fake detection, the authors included additional 90K unedited images; and 3) the authors introduced region-based image-level classification accuracy as an additional assessment criterion. Interesting and important work. Enjoyed reading.

**Strengths:**

see the above summary.

**Weaknesses:**

Two points to clarify, rather than weakness.

**Questions:**

Two minor comments/questions: first, in the first paragraph, the authors wrote “the upstream SOTA generative models and their applications, the midstream existing deep fake datasets, as well as the downstream fake detection formulation and models”. I am just curious why the authors define “upstream, midstream, downstream” this way. Also, the authors consider datasets that include humans in this work, and target human face (either the whole face or part of it) manipulation. What about other datasets of natural scenes, animals, etc.? Can the proposed method be applied there as well?

---

> ### Author Response · Authors · 2024-11-22
>
> > **Q1:** In the first paragraph, the authors wrote “the upstream SOTA generative models and their applications, the midstream existing deep fake datasets, as well as the downstream fake detection formulation and models”. I am just curious why the authors define “upstream, midstream, downstream” this way.
>
> **A1:** Thank you for the question. Our categorization is based on the intuitive flow of the research pipeline in this domain. This structure reflects the natural progression in the pipeline: generative models produce synthetic content (upstream), datasets organize and standardize this content (midstream), and detection models apply this data to tackle real-world challenges (downstream). We hope this clarifies the rationale behind our terminology.
>
> ---
>
> > **Q2:** The authors consider datasets that include humans in this work, and target human face (either the whole face or part of it) manipulation. What about other datasets of natural scenes, animals, etc.? Can the proposed method be applied there as well?
>
> **A2:** Yes, our proposed construction method has strong generalization ability and can also be applied to other scenarios, such as natural scenes or animals. The dataset construction pipeline described in Section 3.2 includes pre-processing, regional editing, and post-rendering, all of which are adaptable to different contexts. In pre-processing, the regions to be edited are specified, which can be done manually or through segmentation methods like SAM. Regional editing can be performed using a generative model suitable for the specific type of scene, and the post-processing step is not constrained by the nature of the input.
>
> That said, we focus on datasets involving human faces in this work because the manipulation of human images, particularly faces, often has the most harmful and urgent societal implications. By targeting this critical area, we aim to address one of the most pressing challenges in the realm of generative model misuse and detection.

---

### Official Review · Reviewer_W2Af · 2024-11-04

**Soundness:** 3
**Presentation:** 3
**Contribution:** 2
**Rating:** 3
**Confidence:** 5

**Summary:**

The paper introduces a new dataset, which includes different and new deepfake tasks like inpainting. It includes the images edited by two GANs and two Diffusion models. The size of the dataset is 30,000. Through human studies and GPT-4 evaluation, the paper shows the difficulty of prediction and detection over DETER by human beings and GPT-4. Through further benchmarking, the paper shows how incorporating unseen and unaltered images improves model performance. It also delivers insights like the regions edited by inpainting is counterintuitively hard to predict.

**Strengths:**

1. The paper presents a new dataset of 30,000 images, which is larger and incorporates different granularities for image editing. It also includes masks for more accurate evaluations.
2. The paper includes many different models for predictions and detection.

**Weaknesses:**

1. In the user study, it is a bad idea to include "I'm not sure" as an option, which will greatly harm the data quality and amount of information. Instead, binary selections can still be enforced, and you can ask participants to indicate their confidence scores. The same is true for GPT-4.
2. The dataset only includes edited images from four different models, which might be insufficient for a comprehensive dataset, especially considering the current flourishing development of generative AI technologies. The claims of difficulty of predicting regions edited by GANs over Diffusion Models are not well supported. By the way, how many models do other SOTA datasets include, as listed in Table 1?
3. A cutoff of IoU of 0.5 is not very informative in showing the benchmarked performance. Why not show the averaged IoU directly?
4. Many of the insights obtained from Section 4.3 are not very informative and are limited to observations. For example, why are images edited by GANs more difficult to predict? Why do eyes and eyebrows raise more errors? Why do models trained on DETER transfer better to OpenForensics (noted that both are face-swapping)?
5. The paper does not include instruction-based image editing, especially those conducted by commercial platforms. This is, however, probably the most significant threat when considering edited images as deepfakes, which can be used and accessed by any ordinary user on the Internet.
6. The dataset is not yet open-sourced. Nor does the paper mention any plans to open-source the dataset.
7. The biggest novelty of the dataset is that it includes different granularities. However, there are no insights into granularities in human studies. For example, why is DETER harder for human beings to identify? Among FaceSwap, Attribute, and Inpaint, which one is the most difficult? It seems that the only purpose of the human study is to show that "it is hard for human eyes".
8. Although images including "multiple faces" are considered novel, no insights are given about this, either in human studies or the benchmark.

**Questions:**

1. In your training setup, to address the issue where “models tend to assume target regions exist in every image,” why didn’t you just use the original, unaltered fake images from the open-source dataset?
2. Considering "another 140K unseen unmanipulated images" in the improved setting, will they be released to the research community?
3. It would be interesting to benchmark AIGC detectors pre-trained on previous SOTA datasets against the DETER dataset.
4. Model performance appears similar across various models, from the 2015 Faster R-CNN to DINO. Does model capacity influence performance in this case, and if not, why?
5. What instructions were fed to GPT-4 for prompting in Section 3.3?
6. Other questions are already mentioned in Weakness.

**Details Of Ethics Concerns:**

The paper proposes a new deepfake dataset that includes human faces.

---

> ### Author Response · Authors · 2024-11-22
>
> > **Q1:** In the user study, it is a bad idea to include "I'm not sure" as an option, which will greatly harm the data quality and amount of information. Instead, binary selections can still be enforced, and you can ask participants to indicate their confidence scores. The same is true for GPT-4.
>
> **A1:** Thank you for this suggestion. When a user is unable to determine the authenticity of an image, it indicates that the image is either genuinely real or very close to appearing real. Forcing a binary “yes or no” decision in such cases could introduce noise into the evaluation, and including an "I'm not sure" option allows us to capture this ambiguity, which is valuable for understanding the limitations of human perception in distinguishing real from fake images.
>
> Typical instantiations of Likert scale ratings often include an odd number of options, with the middle one representing neutrality, such as “neither agree nor disagree.” In our case, this middle option corresponds to “both images look similarly real,” which aligns with our evaluation objectives.
>
> ---
>
> > **Q2:** The dataset only includes edited images from four different models, which might be insufficient for a comprehensive dataset, especially considering the current flourishing development of generative AI technologies.
>
> **A2:** Thank you for raising this important point. We address your concern from two perspectives:
>
> **Selection of generative methods:** While there is indeed a flourishing development of generative AI technologies, not all models are designed to generate highly realistic human images, which is the focus of our dataset for deepfake detection modeling community. Our goal is to select models that best support the construction of a high-quality dataset for this specific task. We experimented with multiple (more than 20) methods during development, partially illustrated in Figure 3 of the main text, and chose the ones that performed most effectively in producing realistic and diverse human faces, and which would thus pose the most challenge for deepfake detection..
>
> **Flexibility of construction pipeline:** Our dataset construction pipeline is designed to be flexible and adaptable, making it straightforward to incorporate newer generative models as they emerge. If more powerful models become available, they can be easily integrated into the pipeline to enhance the dataset’s diversity and utility.
>
> Additionally, we will release the construction code as part of our work. We appreciate this valuable feedback and agree on the importance of keeping updated with advancements in generative modeling. As part of our maintenance plan, we intend to regularly update the testing set to reflect emerging methods, encouraging the generalizability of detection models trained on DETER against more diverse generators.
>
> ---
>
> > **Q3:** The claims of difficulty of predicting regions edited by GANs over Diffusion Models are not well supported.
>
> **A3:** As we wrote in the Line 511-513: “We observe that detection models trained with the GANs-based generators can generalize well to the DMs-based testing images, while the inverse setting induces a non-trivial performance drop”. Supported with the experimental results in Table 6 (right), where the model trained with GAN-based generators achieved an AP of 0.87 on GAN-based and 0.88 on DM-based test images, while the model trained with DM-based generators achieved an AP of 0.71 on GAN-based and 0.89 on DM-based test images, respectively. This is **an empirical yet valid observation under the current task scenario**, and we do not believe it constitutes an overclaim or misconception.
>
> Since this work is positioned as a dataset and benchmark paper, we believe this finding could be of interest to readers in its current context. While a deeper investigation into this particular research question from a broader or theoretical perspective would indeed be valuable, we consider it to be beyond the scope of this work.

---

> > ### Author Response · Authors · 2024-11-22
> >
> > > **Q4:** How many models do other SOTA datasets include, as listed in Table 1?
> >
> > **A4:** We list the number of sources for other datasets below. While it is possible to include a larger number of methods, we emphasize that simply adding more methods without proper quality control **does not necessarily improve the dataset’s utility.** The quality of the data is critical for advancing detection models. Including "easy examples" in the dataset may lead to artificially inflated detection scores, giving the appearance of improved performance. However, this does not translate to better generalizability or robustness in detecting true fake images. Our approach prioritizes selecting high-quality methods that contribute to meaningful advancements in detection model capabilities rather than simply increasing the dataset size.
> >
> > |       Method        | Number |      Method       | Number |
> > | :-----------------: | :----: | :---------------: | :----: |
> > | FaceForensics++ 19’ |   5    | OpenForensics 21’ |   1    |
> > |    Celeb-DF 20’     |   1    |  DF-Platter 23’   |   3    |
> > |      DFFD 20’       |   7    |     DGM4 23’      |   4    |
> > |      DFDC 20’       |   8    | DETER 24’ (Ours)  |   4    |
> > |   ForgeryNet 21’    |   15   |                   |        |
> >
> > ---
> >
> > > **Q5:** A cutoff of IoU of 0.5 is not very informative in showing the benchmarked performance. Why not show the averaged IoU directly?
> >
> > **A5:** Thank you for this question. In our work, we draw inspiration from common practices in computer vision tasks, such as pedestrian detection [1] and referring expression comprehension [2], where an IoU threshold of 0.5 is often used to evaluate localization accuracy. This metric serves as a standardized and reasonable way to demonstrate benchmark performance in many scenarios.
> >
> > The choice of IoU = 0.5 (Average Precision) emphasizes the recall rate for region localization, whereas the average IoU metric focuses on how well the model fits the boundaries of fake regions. Our goal with IoU > 0.5 is to determine whether the fake region is correctly detected, which aligns with real-world applications where high recall is prioritized. Blindly optimizing for a high average IoU could lead to models focusing excessively on boundary fitting, potentially compromising recall.
> >
> > **Additionally, average IoU can be disproportionately influenced by the size of artifacts.** Large fake regions contribute more to the IoU score than multiple smaller regions, which is not always desirable. For instance, a model that perfectly detects a single large fake region (100x100) but misses nine smaller regions (10x10 each) could still achieve a high average IoU (100x100 / (100x100 + 9x10x10) = 91.7%), even though the recall in this case would only be 0.10. Using IoU = 0.5 as the threshold avoids such distortions and ensures a balanced evaluation of performance.
> >
> > [1] Dollar, Piotr, et al. "Pedestrian detection: An evaluation of the state of the art." TPAMI 2021. (citation 4138)
> >
> > [2] Yu, Licheng, et al. "Mattnet: Modular attention network for referring expression comprehension." CVPR. 2018. (citation 894)
> >
> > ---
> >
> > > **Q6:** Many of the insights obtained from Sec 4.3 are not very informative. For example, (1) why are images edited by GANs more difficult to predict? (2) Why do eyes and eyebrows raise more errors? (3) Why do models trained on DETER transfer better to OpenForensics (noted that both are face-swapping)?
> >
> > **A6:** Thank you for the question.
> >
> > (1) One potential reason is the difference in how diffusion-based and GAN-based methods approach the inpainting task. Diffusion-based methods are not natively designed for inpainting tasks, but leverage their generative capabilities to reconstruct full images for masked region prediction, often without specifically optimizing for the masked region. In contrast, GAN-based methods are explicitly designed with specialized decoders that focus on accurately predicting the masked regions. This targeted optimization allows GAN-based methods to produce higher-quality and more seamless edits, leaving subtler traces. As a result, detecting edits in images generated by GAN-based methods is inherently more challenging.
> >
> > (2) As shown in Figure 5, there is a distribution difference in the ground truth of the **attribute editing task**, with higher frequencies for the eye and eyebrow attributes. Compared to Mask2Former, YOLACT amplifies the frequency differences in the dataset more, causing it to focus more on errors around the eyes and eyebrows.
> >
> > (3) The editing traces in our dataset are more subtle, and the quality of the generated images is higher, enabling models trained on DETER to learn to detect more nuanced false cues. Additionally, it is generally easier for models to generalize to datasets with more apparent artifacts, like OpenForensics, when they are trained on harder cases with more subtle traces. This observation highlights the quality of our dataset in fostering robust and transferable detection capabilities.

---

> > > ### Author Response · Authors · 2024-11-22
> > >
> > > > **Q7:** The paper does not include instruction-based image editing, especially those conducted by commercial platforms. This is, however, probably the most significant threat when considering edited images as deepfakes, which can be used and accessed by any ordinary user on the Internet.
> > >
> > > **A7:** The models we selected require a mask input for the desired manipulation area, with the mask also serving as the ground truth. Existing instruction-based image editing methods [1] alter the entire image without specifying the manipulated area, which doesn't meet our objectives. Meanwhile, it is challenging to design universal and diverse instructions for different images, which significantly limits the scale of the generated dataset. Additionally, we opted for the currently best-performing open-source model for image manipulation, which generates fake images that are on par with commercial models. Due to performance and open-source considerations, we decided against using commercial models.
> > >
> > > [1] Fu, Tsu-Jui, et al. "Guiding instruction-based image editing via multimodal large language models." ICLR 2024.
> > >
> > > ---
> > >
> > > > **Q8:** The dataset is not yet open-sourced. Nor does the paper mention any plans to open-source the dataset.
> > >
> > > **A8:** Thank you for raising this point. We plan to release our dataset upon acceptance. Additionally, we will provide the code for constructing the pipeline, including implementation details for different granularities of manipulation and post-processing, to facilitate the use of other models for generating fake images.
> > >
> > > ---
> > >
> > > > **Q9:** The biggest novelty of the dataset is that it includes different granularities. However, there are no insights into granularities in human studies. For example, why is DETER harder for human beings to identify? Among FaceSwap, Attribute, and Inpaint, which one is the most difficult? It seems that the only purpose of the human study is to show that "it is hard for human eyes".
> > >
> > > **A9:** In the regional fake detection task, only attribute editing and inpainting images are involved. As shown in the table, both tasks are similarly challenging for humans, as selecting the correctly manipulated region is difficult. The probability of choosing the ground truth (i.e., the region actually manipulated) is around 30%, which is nearly equivalent to random guessing, given that only one out of three regions is the ground truth. This suggests that images of any granularity in our dataset are difficult to identify.
> > >
> > > |                   | Random |  GT   | Unsure | Total |
> > > | :---------------: | :----: | :---: | :----: | :---: |
> > > | Attribute editing | 52.6%  | 30.7% | 16.7%  | 100%  |
> > > |    Inpainting     | 61.3%  | 30.1% |  8.6%  | 100%  |
> > >
> > > ---
> > >
> > > > **Q10:** Although images including "multiple faces" are considered novel, no insights are given about this, either in human studies or the benchmark.
> > >
> > > **A10:** Since multi-face scenarios are common in real-world applications, we included some multi-face images to simulate real scenes. However, for the region-based image-level classification accuracy metric we propose, an image is considered correctly classified only when there are no missed detections or false positives. The multi-face case is significantly more difficult, and comparing the performance between multi-face and single-face scenarios is not fair. Meanwhile, the former can be viewed as a combination of multiple single faces in the regional fake image detection task, so we did not discuss it separately.
> > >
> > > ---
> > >
> > > > **Q11 & Q12:** In your training setup, to address the issue where “models tend to assume target regions exist in every image,” why didn’t you just use the original, unaltered fake images from the open-source dataset? Considering "another 140K unseen unmanipulated images" in the improved setting, will they be released to the research community?
> > >
> > > **A11 & A12:** Thank you for pointing this out, and we apologize for any confusion caused by our terminology. To clarify, we have already used the original, unaltered images from the open-source datasets as part of our training setup. The term "unseen" may have been misleading—what we meant to convey was "unmanipulated images from the initial datasets." These images were not altered in our construction pipeline and are already publicly available under the licenses of the original datasets (e.g., CelebA, WiderFace). We will revise the wording in the paper to ensure this is communicated more clearly.

---

> > > > ### Author Response · Authors · 2024-11-22
> > > >
> > > > > **Q13:** It would be interesting to benchmark AIGC detectors pre-trained on previous SOTA datasets against the DETER dataset.
> > > >
> > > > **A13:** Thank you for this suggestion. We believe this is already addressed in our experiments described in Lines 490–494 under **Generalization Ability across Dataset**s\*\*, where we evaluate the performance of detectors pre-trained on the OpenForensics dataset against the DETER dataset. This provides a direct comparison of generalization ability across datasets.
> > > >
> > > > If this does not fully address your question or if there is a specific aspect of benchmarking you would like us to explore further, we would be happy to hear your thoughts and consider additional experiments.
> > > > | Cross Dataset | AP |
> > > > |:----------------------:|:----:|
> > > > | OpenForensics -> DETER | 0.02 |
> > > > | DETER -> OpenForensics | 0.69 |
> > > >
> > > > ---
> > > >
> > > > > **Q14:** Model performance appears similar across various models, from the 2015 Faster R-CNN to DINO. Does model capacity influence performance in this case, and if not, why?
> > > >
> > > > **A14:** The detection model is designed to focus on object localization and category classification, excluding real/fake classification from its intended purpose. Therefore, object detectors are developed to enhance accuracy in locating objects and distinguishing between categories, without targeting real/fake differentiation. Additionally, the fake images in our dataset are highly realistic, making it challenging to distinguish between real and fake image features. Consequently, even with increased detection model capacity, performance gains are minimal. These findings indicate the need for specialized models for this task, rather than directly applying existing methods.
> > > >
> > > > ---
> > > >
> > > > > **Q15:** What instructions were fed to GPT-4 for prompting in Section 3.3?
> > > >
> > > > **A15:** The instructions for GPT-4 prompting are designed similarly to the human study setup in Figure 9 - 12. We first describe the task: “AI technique can edit or create human face images. We will show you three photos, two of those are real photos of humans and one has been edited or created by generative AI. Your task is to select the one coming from AI. You can also choose to say “I am not sure” in case you find it very difficult to identify.” Then we provide an example of an image triplet and provide the correct answer before starting the test.

---

### Comment · Area_Chair_6Guf · 2024-11-24

Dear reviewers,

Thanks for serving as a reviewer. As the discussion period comes to a close and the authors have submitted their rebuttals, I kindly ask you to take a moment to review them and provide any final comments.

If you have already updated your comments, please disregard this message.

Thank you once again for your dedication to the OpenReview process.

Best,

Area Chair

---

### Author Response · Authors · 2024-11-27

Dear Reviewers,

As we approach the deadline of discussion, we would like to address any further questions or concerns you may have regarding our submission. We look forward to engaging in the post-rebuttal discussion!

Best regards,

Authors of DETER

---

### Meta-Review · Area_Chair_6Guf · 2024-12-19

**Metareview:**

The paper introduces a new dataset, including different and new deepfake tasks like image editing or inpainting, edited by GANs and Diffusion models. The size of the dataset is 30,000. The authors also do human studies and GPT-4 evaluation to show the difficulties of predictions and detection over DETER. Including this new dataset can further improve the fake detection.

Strength:

1. The research problem is useful. Datasets for regional deepfake detection is valuable for de-fake.

2. The paper is good writing.

Weaknesses:

1. The dataset size is not large enough and may quickly be out-of-date due to the development of GenAI. Considering many fake image detection papers will generate image datasets by themselves, the real impacts be limited.

2. Lack of evaluations on state-of-the-art fake image detection methods.

3. Possible ethical problems when publishing this dataset.

Most reviews raise their concerns about the paper's impacts, evaluation, analysis, and ethical problems. I think instead of publishing a debatable dataset, exploring the failure reasons of state-of-the-art fake image detection methods may be a better way and will also benefit the community more. Therefore, I tend to reject this paper.

**Additional Comments On Reviewer Discussion:**

Most reviews raise their concerns about the paper's impacts, evaluation, analysis, and ethical problems.

---

### Decision · Program_Chairs · 2025-01-22

Reject